# Stratification of hospitalized COVID-19 patients into clinical severity progression groups by immuno-phenotyping and machine learning

Yvonne M. Mueller [1], Thijs J. Schrama [1], Rik Ruijten[1], Marco W. J. Schreurs[1], Dwin G. B. Grashof [1], Harmen J. G. van de Werken [1,2], Giovanna Jona Lasinio [3], Daniel Álvarez-Sierra [4], Caoimhe H. Kiernan[1], Melisa D. Castro Eiro [1], Marjan van Meurs[1], Inge Brouwers-Haspels [1], Manzhi Zhao[1], Ling Li[1], Harm de Wit[1], Christos A. Ouzounis[5,6], Merel E. P. Wilmsen[1], Tessa M. Alofs[1], Danique A. Laport[1], Tamara van Wees[1], Geoffrey Kraker[7], Maria C. Jaimes[7], Sebastiaan Van Bockstael[7], Manuel Hernández-González[4,8,9], Casper Rokx [10], Bart J. A. Rijnders[10], Ricardo Pujol-Borrell[4,8,9,11] & Peter D. Katsikis [1]✉

Quantitative or qualitative differences in immunity may drive clinical severity in COVID-19. Although longitudinal studies to record the course of immunological changes are ample, they do not necessarily predict clinical progression at the time of hospital admission. Here we show, by a machine learning approach using serum pro-inflammatory, anti-inflammatory and anti-viral cytokine and anti-SARS-CoV-2 antibody measurements as input data, that COVID-19 patients cluster into three distinct immune phenotype groups. These immune-types, determined by unsupervised hierarchical clustering that is agnostic to severity, predict clinical course. The identified immune-types do not associate with disease duration at hospital admittance, but rather reflect variations in the nature and kinetics of individual patient's immune response. Thus, our work provides an immune-type based scheme to stratify COVID-19 patients at hospital admittance into high and low risk clinical categories with distinct cytokine and antibody profiles that may guide personalized therapy.

[1] Department of Immunology, Erasmus University Medical Center, Rotterdam, The Netherlands. [2] Cancer Computational Biology Center, Erasmus MC Cancer Institute, Erasmus University Medical Center, Rotterdam, The Netherlands. [3] Department of Statistical Sciences, University of Rome "La Sapienza", Roma, Italy. [4] Immunology Division, Hospital Universitari Vall d'Hebron, Campus Vall d'Hebron, Barcelona, Spain. [5] School of Informatics, Faculty of Sciences, Aristotle University of Thessaloniki, Thessalonica, Greece. [6] Chemical Process & Energy Resources Institute, Centre for Research & Technology Hellas, Thermi, Thessalonica, Greece. [7] Cytek Biosciences, Fremont, CA, USA. [8] Cell Biology, Physiology and Immunology Department, Universitat Autònoma de Barcelona, Barcelona, Spain. [9] Translational Immunology Research Group, Vall d'Hebron Institut de Recerca (VHIR), Campus Vall d'Hebron, Barcelona, Spain. [10] Department of Internal Medicine, Section Infectious Diseases, and Department of Medical Microbiology and Infectious Diseases, Erasmus University Medical Center, Rotterdam, The Netherlands. [11] Present address: Vall d'Hebron Institute of Oncology (VHIO), Barcelona, Spain. ✉email: p.katsikis@erasmusmc.nl

The newly emerged SARS-CoV-2 virus has caused the COVID-19 pandemic and infected >120 million people over the world, resulting in >2.8 million deaths[1]. In the absence of a highly effective therapy against COVID-19, there remains an urgent need to understand both the pathological mechanisms that lead to severe disease but to also identify clear phenotypes that predict disease severity progression and outcome as this may instruct a more personalized therapy. In an attempt to understand the features of COVID-19 that associate with disease severity, studies have aimed at capturing the perturbation of the immune system and the associated inflammatory syndrome observed. Some of these studies have applied high-dimensional analysis using multiplex cytokines, flow or mass cytometry, or scRNAseq to identify changes in cytokine profiles, peripheral blood immune cell composition and/or gene expression related to COVID-19 severity. Universally, however, these studies have employed disease severity classification to identify immunotypes that characterize mild, moderate or severe disease[2–8]. Although, these studies have identified specific changes present in COVID-19 patients compared with healthy individuals, identifying clear immunotypes that strongly associate with or predict disease severity has proven more challenging[2–5]. Defining, however, immunotypes based on clinical severity is based on the assumption that a single mechanism underlies all patients and that kinetics are exclusively driven by days of infection. This approach is, thus, hampered by the dynamic nature of the immune and inflammatory response to SARS-CoV-2 virus, the very different kinetics that individual patients may exhibit, and the likelihood that very different immune mechanisms underlie the same clinical severity.

By applying machine learning to a discovery and a validation cohort, here we show that COVID-19 patients can be classified, at hospital admittance, into distinct immune-phenotypes. These immunotypes predict subsequent clinical progression and outcome. Such immunotypes can guide the development of practical biomarkers but may also instruct more personalized treatments.

## Results

**Distinct immunotypes are identified by machine learning in acute COVID-19 disease.** In this study, we chose to take an unbiased approach in terms of clinical severity to identify immunotypes by first defining immunotypes in COVID-19 patients and then examining if these relate to clinical severity and progression. At time of hospital entry, we measured in the serum of COVID-19 patients (Rotterdam discovery cohort; $n = 50$, Table 1) modules of specific cytokines with pro-inflammatory, anti-inflammatory or anti-viral activities. We combined these serum cytokines with the host adaptive antibody response and applied machine learning using unsupervised hierarchical clustering to identify immunophenotypes that capture both innate and adaptive responses to SARS-CoV-2 infection. Investigators were not involved in clustering or grouping of patients. Importantly, we did not use clinical severity as a clustering variable. Using this approach, we identified three distinct immunotypes, (labeled: balanced response immunotype: BRI, excessive inflammation immunotype: EXI, and low antibody immunotype: LAI) in acutely infected COVID-19 patients (Fig. 1A). To validate these immunotypes, we applied the same machine learning approach on a second independent cohort of patients from a hospital in Barcelona (Barcelona validation cohort; $n = 88$, Table 1) (Fig. 1B). Principal component analysis (PCA) showed that measurements of the Barcelona cohort data matched very well with the Rotterdam data (Fig. 1C). Independent unsupervised hierarchical clustering of the measurements from the Barcelona cohort consistently revealed a very similar classification of patients into

three distinct immunotypes BRI, EXI, and LAI, who exhibited similar cytokine and antibody characteristics as those discovered in the Rotterdam cohort (Fig. 1A, B).

We further tested the robustness of our immunotypes using both cohorts and a multinomial model. The model showed a high level of accuracy (100%) in predicting the immunotypes using the input data. Next, we estimated the stratification accuracy from a predictive perspective by cross-validation of the dataset into a test and training set. We used increasing test percentages from 5%, 10%, 15%, 20%, and 25% and using 150 iterations. The latter resulted in a median accuracy of 0.83, 0.80, 0.80, 0.80, 0.81, respectively, which shows that the model prediction is marginally depending on the training set size. The above confirmed the robustness of our immunotypes.

The three immunotypes were characterized by distinct serum cytokine profiles and anti-SARS-CoV-2 antibody responses (Fig. 2). Anti-SARS-CoV- nucleocapsid protein (NC) antibodies were measured since this CE-certified ELISA system allowed detection of NC-specific IgM, IgG and IgA isotypes and this could reveal isotype-specific differences. Compared to healthy controls, all three immunotypes had increased pro-inflammatory cytokines (Fig. 2A and Supplementary Fig. S1), displaying further significant differences between them. Immunotype BRI was characterized by low pro-inflammatory, anti-viral and anti-inflammatory cytokines and normal TGFβ1 levels (Fig. 2A and Supplementary Fig. S1). BRI exhibited robust IgM, IgG, and IgA anti-SARS-CoV-2 NC antibodies (Figs. 1A, B and 2B). In contrast, EXI had a much more pro-inflammatory profile, low IFNα and normal TGFβ1 (Fig. 2A). Immunotype EXI was also associated with IgM, IgG and IgA anti-SARS-CoV-2 antibodies (Figs. 1A, B and 2B). Immunotype LAI exhibited a distinct profile from the previous two and was characterized by the presence of a strong IFNα response, reduced TGFβ1 (Fig. 2A), and very low antibody immunity (Figs. 1A, B and 2B). Pro-inflammatory cytokines were significantly higher in EXI compared to both other immunotypes (Fig. 2A). IL-17A was increased in some LAI and EXI patients, whereas IL-5 was decreased in some LAI patients (Supplementary Fig. S1). Very few patients had IL-2 or IL-12 in serum (Supplementary Fig. S1) while IL-4 was undetectable (data not shown). To verify that anti-NC responses is an appropriate surrogate for the overall anti-SARS-Cov-2 antibody response, the anti-NC IgG response was correlated with anti-Spike receptor-binding domain (RBD) Ig response (Supplementary Fig. S2A, total anti-RBD Ig) and serum neutralizing anti-SARS-Cov-2 titers (Supplementary Fig. S2B, PRNT50) in the Rotterdam cohort. Anti-NC IgG responses strongly correlated with both (Supplementary Fig. S2A, B).

As expected, we found that anti-SARS-CoV-2 IgM, IgG and IgA antibodies correlated strongly with each other (Fig. 2C). The strongest cytokine correlation with antibodies, was a negative correlation with IFNα (Fig. 2C). This could indicate that antibodies reduce viral loads and thus IFNα or conversely, high IFNα levels delay or inhibit antibody production. Pro-inflammatory cytokines correlated with each other (Fig. 2C). Anti-viral cytokines IFNα and IFNγ correlated with each other, while IFNα negatively correlated with TGFβ1 (Fig. 2C). Thus, characteristics of the three immunotypes could be driven by distinct cytokine networks in action.

**COVID-19 immunotypes predict clinical severity progression.** We next investigated whether these distinct immunotypes associated with clinical parameters (Table 2). As mentioned, clinical severity was not used as a variable for the unsupervised hierarchical clustering. At study entry, BRI and LAI did not differ in WHO clinical severity scores[9] while EXI was significantly higher

**Table 1 Clinical and laboratory characteristics of Rotterdam discovery and Barcelona validation cohorts.**

| | Level | Overall | Barcelona | Rotterdam |
|---|---|---|---|---|
| N | | 138 | 88 | 50 |
| Immunotype | BRI | 33 (23.9%) | 19 (21.6%) | 14 (28.0%) |
| | EXI | 46 (33.3%) | 33 (37.5%) | 13 (26.0%) |
| | LAI | 59 (42.8%) | 36 (40.9%) | 23 (46.0%) |
| Gender | Male | 91 (65.9%) | 58 (65.9%) | 33 (66.0%) |
| Age | Years | 62 (54–70) | 61 (50–70) | 63 (57.25–69) |
| Days from symptom onset | | 8 (6-12) [$n = 134$] | 8 (6-10) [$n = 84$] | 9 (6-14.75) |
| WHO 8-point score at study entry | 3 | 38 (27.5%) | 34 (38.6%) | 4 (8.0%) |
| | 4 | 60 (43.5%) | 31 (35.2%) | 29 (58.0%) |
| | 5 | 20 (14.5%) | 12 (13.6%) | 8 (16.0%) |
| | 6 | 12 (8.7%) | 10 (11.4%) | 2 (4.0%) |
| | 7 | 8 (5.8%) | 1 (1.1%) | 7 (14.0%) |
| Obesity | Yes | 43 (31.2%) | 20 (22.7%) | 23 (46.0%) |
| | NA | 6 (4.3%) | 2 (2.3%) | 4 (8.0%) |
| Diabetes mellitus | Yes | 32 (23.2%) | 18 (20.5%) | 14 (28.0%) |
| | NA | 2 (1.4%) | 2 (2.3%) | 0 (0.0%) |
| Heart disease | Yes | 48 (34.8%) | 35 (39.8%) | 13 (26.0%) |
| | NA | 2 (1.4%) | 2 (2.3%) | 0 (0.0%) |
| Lung disease | Yes | 30 (21.7%) | 15 (17.0%) | 15 (30.0%) |
| | NA | 2 (1.4%) | 2 (2.3%) | 0 (0.0%) |
| Kidney disease | Yes | 5 (3.6%) | 3 (3.4%) | 2 (4.0%) |
| | NA | 2 (1.4%) | 2 (2.3%) | 0 (0.0%) |
| Liver disease | Yes | 6 (4.3%) | 5 (5.7%) | 1 (2.0%) |
| | NA | 2 (1.4%) | 2 (2.3%) | 0 (0.0%) |
| Cancer | Yes | 7 (5.1%) | 3 (3.4%) | 4 (8.0%) |
| | NA | 2 (1.4%) | 2 (2.3%) | 0 (0.0%) |
| Fever (>38 °C) | Yes | 91 (65.9%) | 68 (77.3%) | 23 (46.0%) |
| | NA | 2 (1.4%) | 2 (2.3%) | 0 (0.0%) |
| C-reactive protein (mg/L) | | 102 (68-181) [$n = 127$] | 116 (73-192) [$n = 78$] | 96 (52-151) [$n = 49$] |
| Ferritin (μg/L) | | 666 (382-1101) [$n = 116$] | 598 (314-1034) [$n = 72$] | 738 (487-1171) [$n = 44$] |
| Lactate dehydrogenase (U/L) | | 369 (293-472) [$n = 121$] | 369 (296-474) [$n = 72$] | 373 (286-461) [$n = 49$] |
| Alanine aminotransferase (U/L) | | 29 (19-54) [$n = 137$] | 32 (18-54) [$n = 88$] | 25 (21-54) [$n = 49$] |
| Bilirubin (μmol/L) | | 9.8 (7.1-13.0) [$n = 122$] | 10.1 (7.8-13.2) [$n = 76$] | 8.5 (6.0-12.0) [$n = 46$] |
| D-dimer (mg/L) | | 0.40 (0.24-0.98) [$n = 120$] | 0.30 (0.19-0.48) [$n = 79$] | 0.97 (0.60-1.49) [$n = 41$] |
| Hemoglobin (mmol/L) | | 8.30 (7.51-9.06) [$n = 136$] | 8.66 (7.88-9.18) [$n = 86$] | 7.65 (6.60-8.30) [$n = 50$] |
| Leukocytes (x$10^9$/L) | | 7.45 (5.90-9.84) [$n = 136$] | 7.62 (6.11-9.87) [$n = 86$] | 7.00 (5.20-9.78) [$n = 50$] |
| Neutrophils (x$10^9$/L) | | 5.70 (4.16-8.01) [$n = 133$] | 6.02 (4.60-8.03) [$n = 86$] | 5.10 (3.70-7.70) [$n = 47$] |
| Lymphocytes (x$10^9$/L) | | 1.02 (0.83-1.41) [$n = 133$] | 1.02 (0.83-1.38) [$n = 86$] | 1.00 (0.80-1.50) [$n = 47$] |
| Monocytes (x$10^9$/L) | | 0.45 (0.30-0.64) [$n = 133$] | 0.48 (0.33-0.66) [$n = 86$] | 0.40 (0.29-0.60) [$n = 47$] |
| Thrombocytes (x$10^9$/L) | | 216 (163-294) [$n = 135$] | 209 (166-289) [$n = 86$] | 229 (151-298) [$n = 49$] |
| Tocilizumab before study entry | Yes | 7 (5.1%) | 0 (0.0%) | 7 (14.0%) |
| | NA | 0 (0.0%) | 0 (0.0%) | 0 (0.0%) |
| Tocilizumab after study entry | Yes | 2 (1.4%) | 0 (0.0%) | 2 (4.0%) |
| | NA | 88 (63.8%) | 88 (100.0%) | 0 (0.0%) |
| Corticosteroids after study entry | Yes | 10 (7.2%) | 0 (0.0%) | 10 (20.0%) |
| | NA | 88 (63.8%) | 88 (100.0%) | 0 (0.0%) |
| Convalescent plasma after study entry | Yes | 29 (21.0%) | 0 (0.0%) | 29 (58.0%) |
| | NA | 88 (63.8%) | 88 (100.0%) | 0 (0.0%) |

Data are n (%) or median (Q1–Q3). Complete data was available for the continuous values shown if not stated otherwise [$n = $]; NA not available.
The clinical characteristics and laboratory measurements of Rotterdam discovery cohort, Barcelona validation cohort and the combination of both shown.

(Fig. 3A). Thus at study entry, the immunotypes are not determined by disease severity. To assess clinical severity progression during hospitalization, we examined the highest/worst clinical score that patients exhibited within 30 days of admittance. During hospitalization, EXI and LAI were characterized by clinical deterioration and higher peak WHO clinical severity scores (median peak score of 6 for both) (Fig. 3A). In contrast, BRI improved after entry and clinical scores declined (the median peak score of 3, was the score of entry) (Fig. 3A). Reflecting the more severe disease scores during hospitalization, all mortality occurred in EXI and LAI patients, while no patients died in BRI (Figs. 1A, B and 3B). Furthermore, total days in hospital (Fig. 3C) and total days in ICU (Fig. 3D) also differ significantly between patients with the BRI immunotype and patients either in the EXI

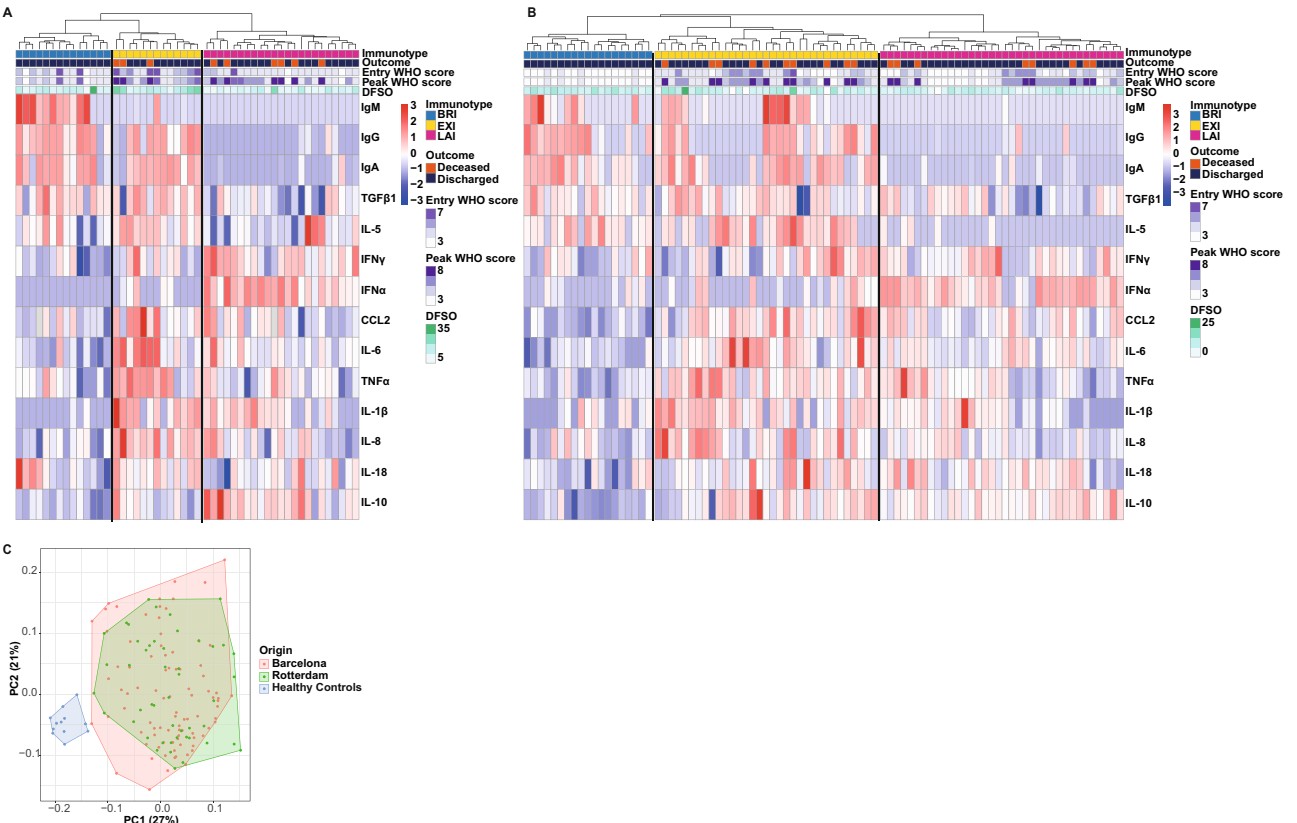

**Fig. 1 Unsupervised hierarchical clustering identifies three distinct immunotypes in acute COVID-19 patients.** Applying machine learning by unsupervised hierarchical clustering solely to serum cytokines and anti-SARS-CoV-2 antibodies identifies three distinct immunotypes. Analysis was performed on samples collected at study entry and without clinical data input. The three immunotypes, identified independently in two patient cohorts, are depicted in **A** the heat map of the Rotterdam discovery cohort (n = 50) and **B** the heat map of the Barcelona validation cohort (n = 88) using row-based log-transformed z-scores. **C** Principal component analysis (PCA) of serum cytokines and anti-SARS-CoV-2 antibodies shows that both Rotterdam and Barcelona cohorts cluster together while they lie apart from healthy controls. The first two components with their percentage of variance are shown in parentheses. Red-blue color depicts z-scores. Top banners of heatmaps show: immunotypes BRI, EXI and LAI, WHO clinical score at entry and the peak during hospitalization, patient death and discharge, and days from symptoms onset (DFSO).

or LAI immunotype group. These differences in severity between immunotypes could not be attributed to age as these did not vary significantly (Fig. 3E). Although sex distribution significantly differs between the BRI and EXI group (Fig. 3F), no significant difference was detected between BRI and LAI and EXI and LAI, indicating that gender is not the predictive determinant of the three immunotypes. To assess and emphasize the importance of the distinct molecular markers and other variables, these variables were directly compared within the three immunotypes using Wilcoxon rank-sum/Mann–Whitney U-test. We additionally applied a logistic regression model to each biomarker using both a Uni- and Multivariate approach with gender, age, and DFSO as additional variables for the multivariate analysis. Overall, no striking difference between the non-parametric and the parametric tests were detected (Supplementary Table S1) and further confirmed that gender, age, and DFSO were not defining parameters of the immunotypes.

Thus, despite immunotypes having a relatively mild disease at study entry, the EXI and LAI phenotypes captured patients that would clinically deteriorate after hospitalization with higher mortality, longer hospital stays and increased ICU days, while BRI identified patients that would improve clinically.

**COVID-19 immunotypes are not defined by disease duration.** Although the three immunotypes differ at time of entry in the number of days from symptoms onset (DFSO), with disease

duration shortest in LAI, differences in DFSO did not explain these phenotypes (Fig. 4A–C). Cytokine and antibody levels did not significantly correlate with DFSO, suggesting that immunotypes are not determined by disease duration (Fig. 4B, C). More importantly, the kinetics of antibodies and cytokines differed between the three immunotypes with LAI even after DFSO of 10 days still having high levels of IFNα while the antibody responses remained muted (Fig. 4B, C). IL-6 and TNFα, on the other hand, were high in EXI patients already at DFSO of 5 days (Fig. 4B). For the individual immunotypes, anti-SARS-CoV-2 antibody responses had different trajectories in terms of DFSO. In BRI and EXI patients, antibodies come up early and stay up (Fig. 4C). The frequency of patients positive for anti-SARS-CoV-2 antibodies was increased in BRI and EXI compared to LAI irrespective of time since symptoms onset (Fig. 4D). Thus, duration of infection could not explain LAI's high IFNα or low antibodies. DFSO also failed to correlate with days of hospitalization and days in ICU (Rs = −0.17, p = 0.08 and Rs = −0.18, p = 0.07, respectively). This supports the notion that immunotypes reflect the individual patient's nature of the response and the rate of development of immunity rather than a strictly linear chronological relationship to duration of infection.

**Immunotypes differ in their clinical laboratory characteristics.** The three immunotypes differed significantly in terms of plasma inflammation markers and blood cell numbers (Table 2). Plasma

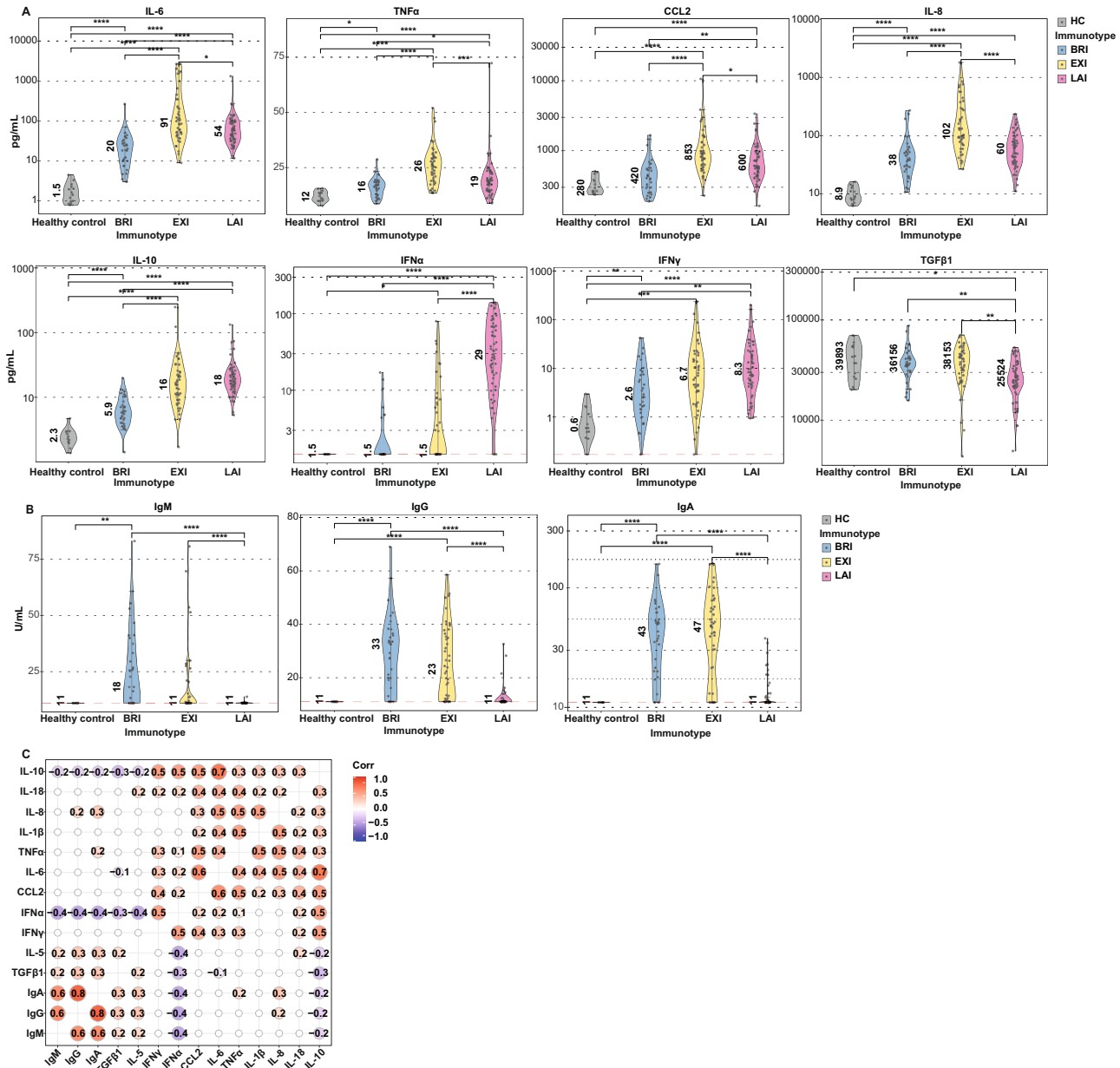

**Fig. 2 COVID-19 immunotypes have distinct cytokine and anti-SARS-CoV-2 antibody levels.** Immunotypes BRI, EXI and LAI have distinct patterns of cytokines and anti-SARS-CoV-2 antibodies. **A** Serum cytokines and **B** IgM, IgG, and IgA anti-SARS-CoV-2 antibodies are shown for BRI ($n = 33$), EXI ($n = 46$) and LAI patients ($n = 59$) and healthy controls ($n = 14$). Violin plots with single data points depict the range of levels and number indicates medians. Whiskers indicate the 1.5x interquartile range (IQR) value. A two-sided Wilcoxon rank-sum statistical test with Bonferroni correction was applied on all six comparisons but only shown for significant differences. Red dashed line indicates lower limit of detection. **C** Correlograms depicting correlations between cytokines and antibodies ($n = 138$ patients). Red indicates positive and blue negative correlation. Only statistically significant association ($p < 0.05$) are shown. Spearman's rho ($R_s$) is shown in circles. *$p < 0.05$, **$p < 0.01$, ***$p < 0.001$, and ****$p < 0.0001$.

levels of markers of inflammation C-reactive protein (CRP), ferritin, d-dimer, Lactate dehydrogenase (LDH) were all significantly higher in EXI (Fig. 5A). Strikingly, these were not increased in LAI compared to BRI. Leukocytes and neutrophils were highest in EXI, while lymphocyte levels trended to be lower in EXI and LAI but were not significant (Fig. 5B). Thrombocytes, however, were significantly lower in LAI patients (Fig. 5B and Table 2). Viral loads in nasopharyngeal swabs were not significantly different between immunotypes but trended to be higher in LAI (Fig. 5C), while within LAI patients, viral loads did not correlate with IFNα levels (Fig. 5D). Finally, we found no correlation between viral load and DFSO (Rs = −0.18, $p = 0.34$).

**Deep phenotyping of immunotypes reveals distinct cellular changes in blood of COVID-19 patients.** To acquire insight into the potential mechanisms that are behind the described three immunotypes we used 40-color spectral flow cytometry and identified the immune cell subset changes associated with these immunotypes (Fig. 6). Although COVID-19 patients as a whole differed clearly from healthy individuals, comparing immunotypes at study entry found remarkably few differences between them. COVID-19 patients compared to healthy individuals had increased plasmablasts (Fig. 6B–D and Supplementary Table S2). The pro-inflammatory intermediate monocytes 1 and 2[10] and pro-inflammatory IgD-negative non-conventional memory B cells

**Table 2 Clinical and laboratory characteristics of BRI, EXI and LAI immunotypes.**

| | Level | BRI | EXI | LAI | *p*-value |
|---|---|---|---|---|---|
| N | | 33 | 46 | 59 | |
| Cohort | Barcelona | 19 (57.6%) | 33 (71.7%) | 36 (61.0%) | 0.367 |
| | Rotterdam | 14 (42.4%) | 13 (28.3%) | 23 (39.0%) | |
| Gender | Male | 15 (45.5%) | 36 (78.3%) | 40 (67.8%) | 0.009 |
| Age | Years | 57 (49–64) | 63 (57–69.75) | 65 (52.50–71) | 0.059 |
| Days from symptom onset | | 10 (8–13) | 10 (6.50–14) [n = 43] | 7 (5–10) [n = 58] | 0.002 |
| WHO 8-point score at study entry | 3 | 17 (51.5%) | 5 (10.9%) | 16 (27.1%) | 0.002 |
| | 4 | 11 (33.3%) | 20 (43.5%) | 29 (49.2%) | |
| | 5 | 3 (9.1%) | 8 (17.4%) | 9 (15.3%) | |
| | 6 | 0 (0.0%) | 8 (17.4%) | 4 (6.8%) | |
| | 7 | 2 (6.1%) | 5 (10.9%) | 1 (1.7%) | |
| Outcome | Deceased | 0 (0.0%) | 13 (28.3%) | 14 (23.7%) | 0.004 |
| | Discharged | 33 (100%) | 33 (71.7%) | 45 (76.3%) | |
| Obesity | Yes | 6 (18.2%) | 13 (28.3%) | 24 (40.7%) | 0.192 |
| | NA | 1 (3.0%) | 3 (6.5%) | 2 (3.4%) | |
| Diabetes mellitus | Yes | 4 (12.1%) | 11 (23.9%) | 17 (28.8%) | 0.113 |
| | NA | 0 (0.0%) | 2 (4.3%) | 0 (0.0%) | |
| Heart disease | Yes | 6 (18.2%) | 20 (43.5%) | 22 (37.3%) | 0.034 |
| | NA | 0 (0.0%) | 2 (4.3%) | 0 (0.0%) | |
| Lung disease | Yes | 8 (24.2%) | 11 (23.9%) | 11 (18.6%) | 0.311 |
| | NA | 0 (0.0%) | 2 (4.3%) | 0 (0.0%) | |
| Kidney disease | Yes | 0 (0.0%) | 4 (8.7%) | 1 (1.7%) | 0.049 |
| | NA | 0 (0.0%) | 2 (4.3%) | 0 (0.0%) | |
| Liver disease | Yes | 1 (3.0%) | 3 (6.5%) | 2 (3.4%) | 0.293 |
| | NA | 0 (0.0%) | 2 (4.3%) | 0 (0.0%) | |
| Cancer | Yes | 2 (6.1%) | 1 (2.2%) | 4 (6.8%) | 0.271 |
| | NA | 0 (0.0%) | 2 (4.3%) | 0 (0.0%) | |
| Fever (>38 °C) | Yes | 15 (45.5%) | 29 (63.0%) | 47 (79.7%) | 0.004 |
| | NA | 0 (0.0%) | 2 (4.3%) | 0 (0.0%) | |
| C-reactive protein (mg/L) | | 82 (39–125) [n = 32] | 159 (86–245) [n = 44] | 94 (59–152) [n = 51] | <0.001 |
| Ferritin (µg/L) | | 434 (278–676) [n = 32] | 887 (648–1883) [n = 38] | 605 (313–1052) [n = 46] | <0.001 |
| Lactate dehydrogenase (U/L) | | 320 (257–381) [n = 32] | 453 (362–599) [n = 39] | 341 (296–454) [n = 50] | <0.001 |
| Alanine aminotransferase (U/L) | | 25 (18–54) | 31 (18–57) [n = 46] | 29 (21–53) [n = 58] | 0.725 |
| Bilirubin (µmol/L) | | 8.9 (6.9–13.0) [n = 31] | 11.4 (8.0–14.8) [n = 44] | 9.0 (6.0–12.2) [n = 47] | 0.1 |
| D-dimer (mg/L) | | 0.39 (0.30–0.61) [n = 29] | 0.89 (0.32–2.22) [n = 39] | 0.30 (0.18–0.62) [n = 52] | <0.001 |
| Hemoglobin (mmol/L) | | 8.01 (7.45–8.69) | 8.30 (7.60–9.05) [n = 44] | 8.38 (7.70–9.12) | 0.352 |
| Leukocytes (x10⁹/L) | | 7.35 (6.10–9.30) | 9.70 (7.10–12.33) [n = 44] | 6.50 (4.84–8.06) | <0.001 |
| Neutrophils (x10⁹/L) | | 5.11 (4.16–7.69) | 7.78 (5.69–10.05) [n = 44] | 4.78 (3.61–6.47) [n = 56] | <0.001 |
| Lymphocytes (x10⁹/L) | | 1.40 (0.90–1.50) | 0.99 (0.82–1.26) [n = 44] | 1.00 (0.80–1.28) [n = 56] | 0.067 |
| Monocytes (x10⁹/L) | | 0.48 (0.40–0.68) | 0.50 (0.30–0.72) [n = 44] | 0.38 (0.27–0.58) [n = 56] | 0.008 |
| Thrombocytes (x10⁹/L) | | 280 (208–343) | 274 (176–320) [n = 44] | 182 (152–229) [n = 58] | <0.001 |
| Tocilizumab before study entry | Yes | 0 (0.0%) | 6 (13.0%) | 1 (1.7%) | 0.01 |
| | NA | 0 (0.0%) | 0 (0.0%) | 0 (0.0%) | |
| Tocilizumab after study entry | Yes | 1 (3.0%) | 0 (0.0%) | 1 (1.7%) | 0.582 |
| | NA | 19 (57.6%) | 33 (71.7%) | 36 (61.0%) | |
| Corticosteroids after study entry | Yes | 3 (9.1%) | 4 (8.7%) | 3 (5.1%) | 0.476 |
| | NA | 19 (57.6%) | 33 (71.7%) | 36 (61.0%) | |
| Convalescent plasma after study entry | Yes | 9 (27.3%) | 5 (10.9%) | 15 (25.4%) | 0.356 |
| | NA | 19 (57.6%) | 33 (71.7%) | 36 (61.0%) | |

Data are *n* (%) or median (Q1–Q3). Complete data was available for the continuous values shown if not stated otherwise [n = ]; *NA* not available. Significance between immunotypes for categorical variables determined by Chi-squared test and for continuous variables by Kruskal–Wallis Rank-Sum Test.
The clinical characteristics and laboratory measurements of patients in each immunotype group are presented.

(IgD- CD27- B cells)[11] were both increased in blood of patients (Fig. 6C; Supplementary Fig. S3 and Supplementary Table S2). In contrast, non-classical monocytes were reduced in blood of patients (Fig. 6C; Supplementary Fig. S3, Supplementary Table S2), and loss of these anti-inflammatory cells may further contribute to immune activation[12,13]. Plasmacytoid dendritic cells (DC) and conventional DC were also both reduced in patients (Fig. 6B–D and Supplementary Table S2). When the three immunotypes were compared with each other, remarkably few differences existed (Fig. 6E, F and Supplementary Table S2). Only intermediate monocytes 2 were

strongly increased in EXI compared to LAI (Fig. 6F, G). Overall, despite the large differences between COVID-19 patients and healthy controls, the differences between immunotypes were subtle, underscoring a disconnect between peripheral blood cells and systemic plasma cytokines in the immunotypes.

## Discussion

Our study does not aim to determine whether COVID-19 differs from other respiratory infections but seeks to identify groups of

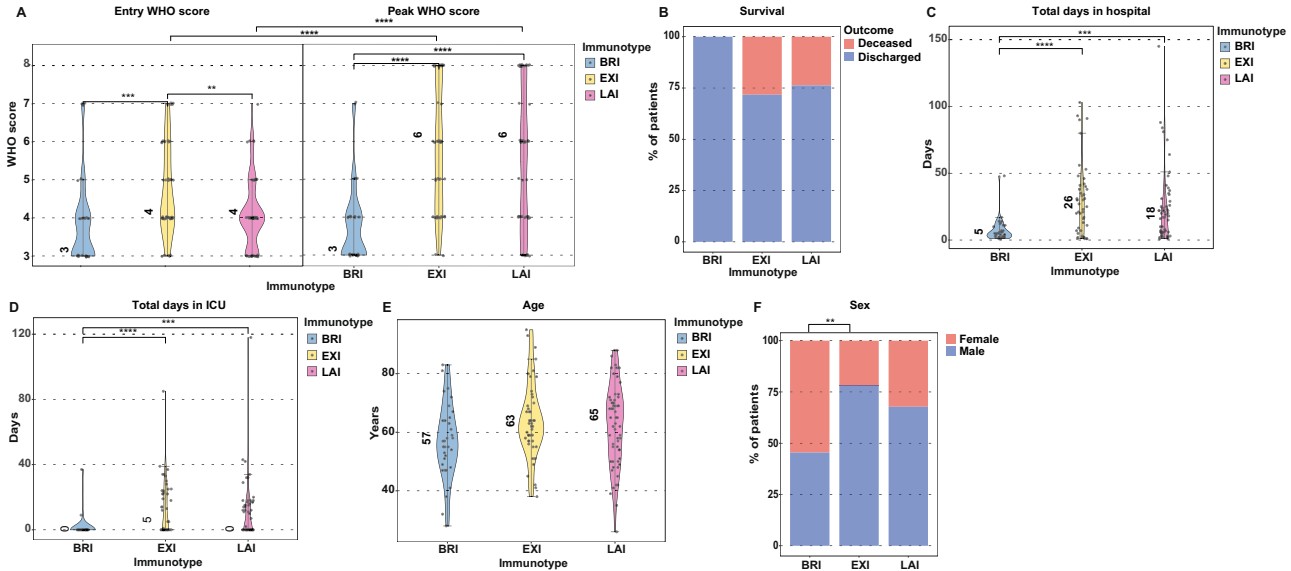

**Fig. 3 Immunotypes predict clinical improvement or deterioration in acute COVID-19.** COVID-19 immunotype BRI identifies at study entry patients that will clinically improve while immunotypes EXI and LAI predict clinical deterioration as peak severity scores during hospitalization are higher than entry scores. Violin plots and stacked bar graphs show per immunotype **A** WHO clinical score at study entry and peak WHO clinical score during hospitalization, **B** Percentage of deceased patients, **C** total days in the hospital, **D** total days in the ICU, **E** Age of patients and **F** Percentage of females and males. Data shown are from BRI ($n = 33$), EXI ($n = 46$), and LAI ($n = 59$) patients. Medians indicated by number in plots. Whiskers depict the 1.5x IQR value. Significance between the three immunotypes for the WHO scores was tested using a two-sided Wilcoxon rank-sum test with Bonferroni correction. Significance for each immunotype between entry WHO score and peak WHO score was tested using a paired Wilcoxon rank-sum test. Significant differences in total days in hospital and ICU were tested using two-sided Wilcoxon rank-sum tests with Bonferroni correction, significant differences in age were tested using two-sided Student's $t$-tests with Bonferroni correction, significance difference in sex distribution was tested using a chi-square test; *$p < 0.05$, **$p < 0.01$, ***$p < 0.001$, and ****$p < 0.0001$.

patients within COVID-19 patients that present a different clinical course. This study was conducted to find phenotypes or identify patient groups and characteristics that can facilitate biomarker discovery, at hospital admission of COVID-19 patients, that predict clinical outcome. These phenotypes or biomarkers could also suggest evidence-based treatment approaches for the patients that will become severely ill. The phenotypes we identified could thus help discover practical biomarkers. This focus on hospital entry is a very different approach compared to other studies performed so far. Previous studies use the approach of grouping COVID-19 patients on the basis of severity to understand mechanisms of disease, which is based on the assumption that one pathophysiological mechanism underlies COVID-19 severity, and that patients respond uniformly to infection and display a similar linear disease progression with time. Using an unbiased analysis that only includes immune parameters can thus provide a more comprehensive mapping of the interplay between the immune response and SARS-CoV-2 virus in patients. Indeed, using this approach, we were able to clearly demarcate immunotypes that predict clinical severity changes and disease outcome. Importantly, the three immunotypes are able to predict disease severity progression, recovery or persistent severe disease. A previous study reported IL-6, IP-10 and IL-10 of first blood draw as predictors of COVID-19 clinical deterioration while antibody responses negatively correlated with deterioration[7]. However, in that study, 25% of patients that clinically improve still have high levels of IP-10 and would be classified as patients that will worsen[7]. Similarly, in our study, IL-6 and IL-10 on their own, do not separate well the three immunotypes as there is considerable overlap in serum levels. Such overlap is also apparent in antibody responses where BRI and EXI have similar anti-SARS-CoV-2 antibody levels but clearly very different clinical courses. Thus, we argue that the

combined signature of multiple cytokines and antibodies has more power to predict disease course and outcomes.

The immunotypes described herein cannot be attributed to strictly chronological differences of the duration of infection in patients. Instead of time since infection, the identified immunotypes reflect the variation in individual patient's kinetics of mounting innate and adaptive immunity to the virus. This notion is supported by the large variation in kinetics with which different hosts mount anti-SARS-CoV-2 immunity[14]. This variation could be due to host factors such as different host genetics[15], epigenetic differences imposed by varying host immunological experiences such as previous pathogen exposures or vaccination[16], or potential microbiome differences[17]. All these factors could lead to varying kinetics of immunity and control of viral loads, which can associate with disease severity and clinical course[18,19]. Non-host related factors such as dose or route of infection may also play a significant role in immunotype development[20]. These elements could all influence the rapidity of mounting innate or adaptive immunity or even the duration of different phases of innate immunity. The question arises, whether there are more immunotypes that can be revealed in larger cohorts of patients. Our finding that small subgroups of patients in our cohorts have increased IL-17A or IL-5 in serum, points to additional immunotypes that may have distinct clinical characteristics in acute infection or associate with post-recovery sequelae of long COVID-19. Our study of 138 patients did not have the power to reveal such rarer immunotypes, but larger cohorts may well be able to do so.

The three identified immunotypes reflect different pathophysiological mechanisms for COVID-19. Based on serum markers of inflammation such as CRP and d-dimer, BRI and LAI are less inflammatory than EXI. However, LAI exhibits more thrombocytopenia compared to BRI and EXI. EXI and LAI had a much

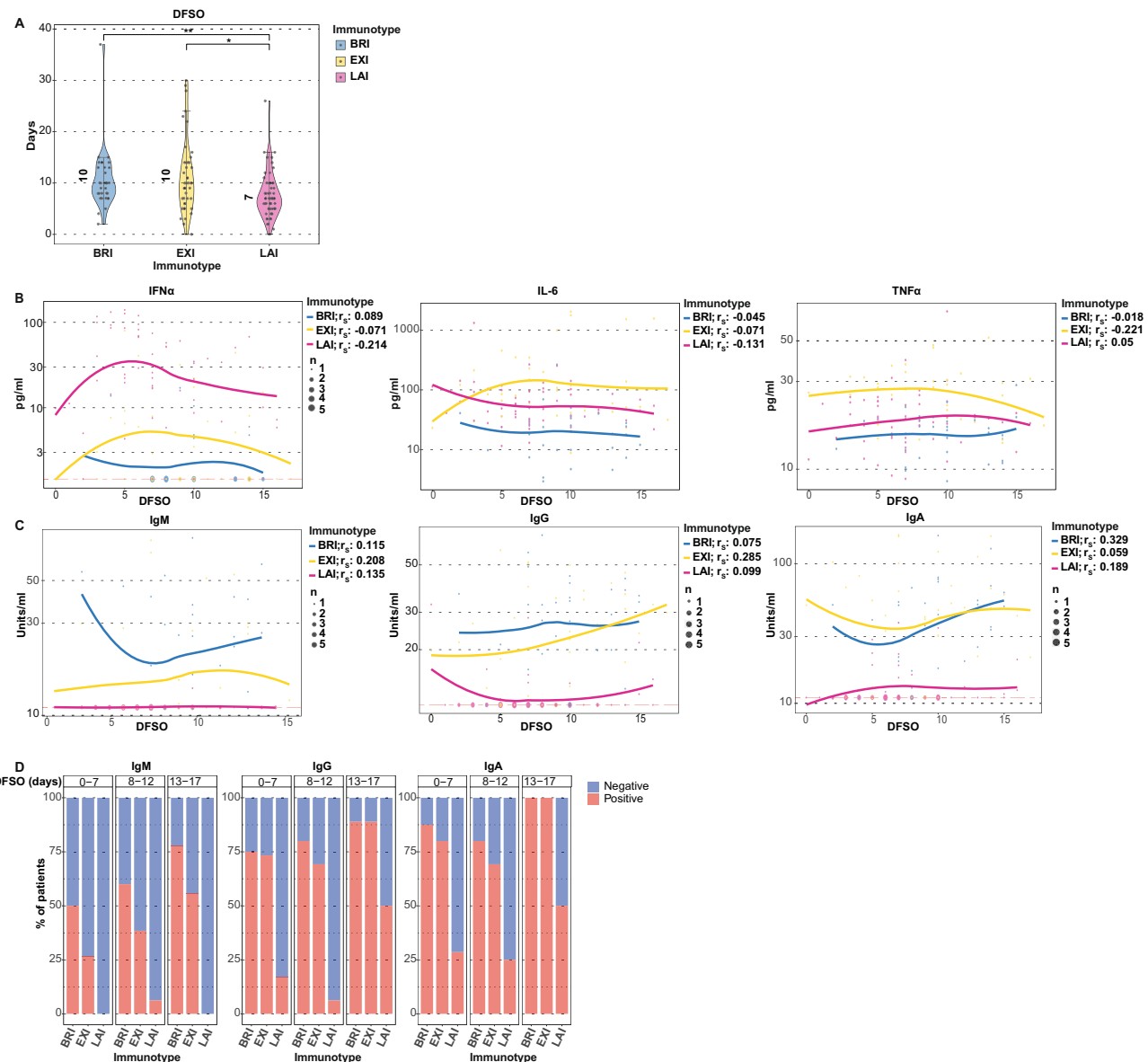

**Fig. 4 Disease duration does not determine immunotypes.** Cytokine differences and the absence of antibodies in LAI are not due to lesser disease duration. **A** Days from symptoms onset (DFSO). Medians shown by number and whiskers indicate the 1.5x IQR value. Two-sided Wilcoxon rank-sum tests with Bonferroni correction were applied. *$p < 0.05$, **$p < 0.01$. **B** Scatterplots of IFNα, IL-6, and TNFα with DFSO per immunotype shown. **C** Scatterplots of anti-SARS-CoV-2 antibodies with DFSO is shown per immunotype. For **B** and **C**, Spearman's rho is shown per immunotype and Bonferroni adjusted p-value is added when significant. Non-parametric local polynomial regression lines plotted using the locally estimated scatterplot smoothing (LOESS) method are shown for individual immunotypes. **D** The percentage of antibody positivity for the three immunotypes is depicted in relation to their DFSO. Red depict percent of patients positive for antibodies, blue depicts percent negative. Data shown are from BRI ($n = 33$), EXI ($n = 46$), and LAI ($n = 59$) patients.

worse clinical course compared to BRI yet the signatures of EXI and LAI differed in terms of their anti-SARS-CoV-2 antibody response, the level of pro-inflammatory cytokines and IFNα while TGFβ1 was uniquely downregulated in LAI. Strikingly, these immunotypes, identified at hospital entry, predicted which patients were less likely to survive, would have clinical severity scores increase, would have longer hospitalizations and would spend more days in the ICU.

Gender is a known factor that can affect clinical severity in COVID-19 with females generally having milder disease. Although females were increased in BRI patients compared to EXI patients, BRI and LAI immunotypes were not statistically different in terms of gender. Univariate and multivariate analysis further excluded that the immunotype differences could be attributed to gender, age or DFSO. We included DFSO in our

analysis to exclude that the different immunotypes we identified, despite being sampled at hospital entry, could be due to some patients being very early in the infection and therefore have not mounted an antibody response yet while other were later in the course of infection. Furthermore, we also confirmed that the immunotypes we identified remain stable over a range of DFSO. Applying a number of different statistical approaches, we can exclude that duration of infection determines immunotypes at hospital admission.

Analysis of the salient association between select pro-inflammatory cytokines, interferons and anti-SARS-CoV-2 antibodies, revealed clear associations of immunotypes with the adaptive immune response, inflammation and the anti-viral response (Fig. 7). BRI appears to mount an early anti-SARS-CoV-2 antibody response that controls viral replication, and dampens

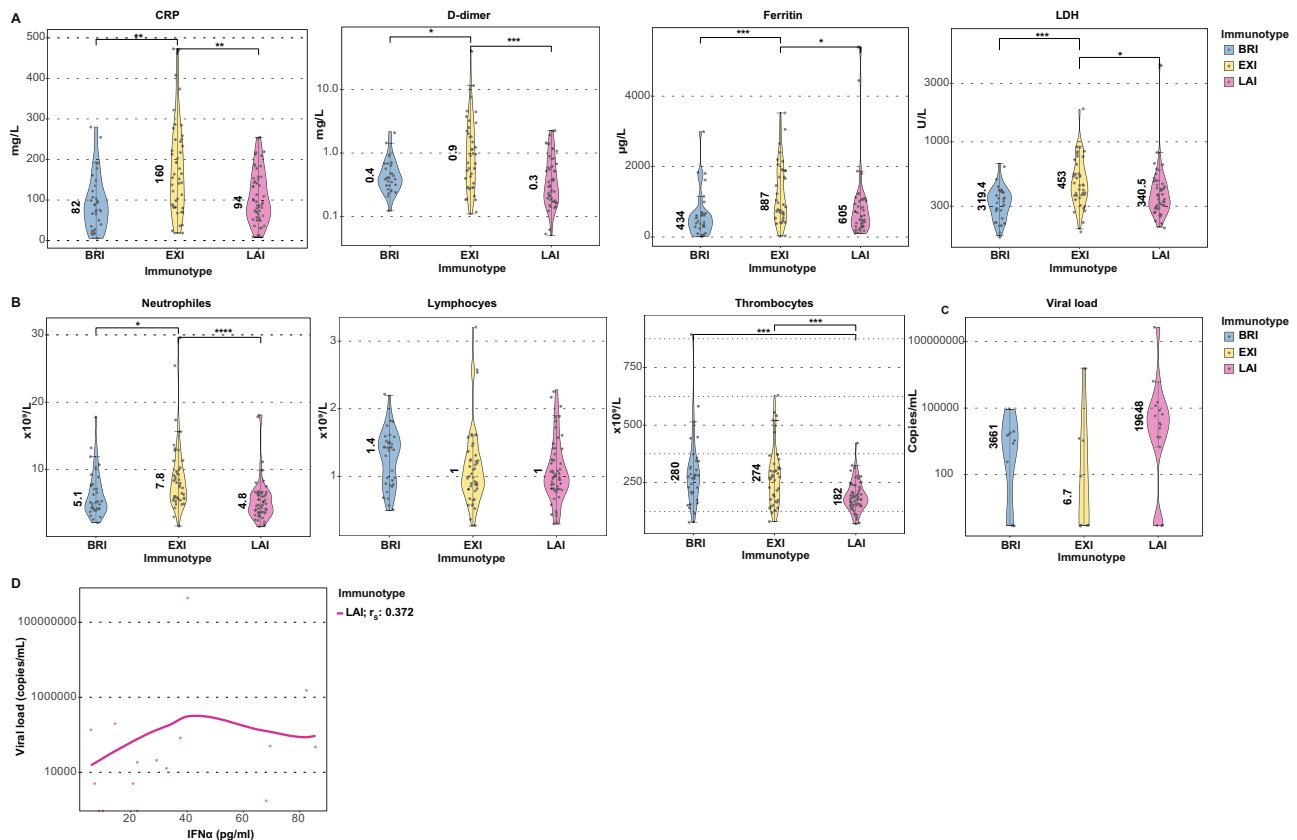

**Fig. 5 Immunotypes differ in their clinical laboratory characteristics. A** Violin plots of CRP, d-dimer, ferritin and LDH measurements shown. **B** Violin plots of blood neutrophil, lymphocyte, and thrombocyte counts. Data from BRI ($n = 33$), EXI ($n = 46$), and LAI ($n = 59$) patients are shown. **C** Viral loads of immunotypes at entry shown (BRI $n = 11$, EXI $n = 8$, and LAI $n = 16$). Medians shown by number in plots. Whiskers indicate the 1.5x IQR value. **D** The scatterplot of IFNα and viral loads in LAI patients shown. Line depicts the non-parametric local polynomial regression line plotted using the locally estimated scatterplot smoothing (LOESS) method. Spearman's rho ($r_s$) is indicated. Correlation was not significant. A subgroup of Rotterdam patients ($n = 35$) with nasopharyngeal viral load measurements are shown in **C** and **D**. Two-sided Wilcoxon rank-sum tests with Bonferroni correction were applied for each measurement in **A**–**C**. *$p < 0.05$, **$p < 0.01$, ***$p < 0.001$, and ****$p < 0.0001$.

the IFN response and subsequent hyper-inflammation. EXI mounts a vigorous inflammatory response. LAI arises as a result of a delayed adaptive immune response against SARS-CoV-2 that leads to sustained high viral loads accompanied by a type I IFN response. Clearly an impaired type I IFN response is detrimental to the clinical course of COVID-19 as indicated by individuals with defects of IFN signaling and antibodies against IFN[21–23]. However, increased type I IFN in serum has also been reported in severe COVID-19 patients[3,24] although others have reported low responses[25]. Such differences may be due to varying patient populations and grouping of different immunotypes. Employing single cell sequencing on just 21 patients, one study reported that anti-SARS-CoV-2 antibodies can hamper type I IFN responses in severe COVID-19 patients[26]. Although this could be the case in EXI patients, such results should be interpreted cautiously, as others have reported that antibodies protect from severe COVID-19[7]. Additionally, we clearly show that high type I IFN and absence of antibodies in LAI patients associates with disease deterioration and markedly worse outcomes, while BRI patients with high antibodies but no type I IFN do well clinically. The increased IFNα and viral loads in LAI, but their failure to correlate (Fig. 5D) indicates that in this immunotype their interaction is potentially deregulated. Such correlations can be seen in vivo in viral infections such as SIV infection, where uncontrolled viremia positively correlates with type I IFN at the chronic phase, yet negatively correlates at early SIV infection[27]. On the other hand, type I IFN can induce ACE2 receptor expression[28].

This can set up a positive feedback loop between viral replication and type I IFN in LAI patients and together with the anti-viral effect of type I IFN may set up a complex non-linear interplay between them.

We used 40-color spectral flow cytometry to identify the immune cell subset changes associated with COVID-19 infection. We find increased plasmablasts as previously described, but also increases in pro-inflammatory cell types such intermediate monocytes 1 and 2, and pro-inflammatory IgD- CD27- non-conventional memory B cells[10,11], while anti-inflammatory cells such as non-classical monocytes were reduced[12,13]. This underscores the tipped balance towards inflammation in patients. Plasmacytoid DC and conventional DC were both reduced in patients and this could impair both anti-viral and adaptive immune responses. Plasmablasts were increased in all three immunotypes irrespective if they had antibodies, arguing they are not a good predictor of anti-SARS-CoV-2 immunity. We found only one difference between immunotypes, an increase in EXI patients of pro-inflammatory intermediate monocytes 2 compared to LAI patients. Thus, COVID-19 patients had significant cellular variations compared to healthy controls, but such differences between immunotypes were not apparent, indicating that the blood cellular compartment does not directly reflect plasma cytokines that may be produced locally in tissue such as lung.

The classification of COVID-19 patients by immunotype may guide personalized therapeutic strategies. Pro-inflammatory cytokines do not distinguish clearly BRI, EXI and LAI from

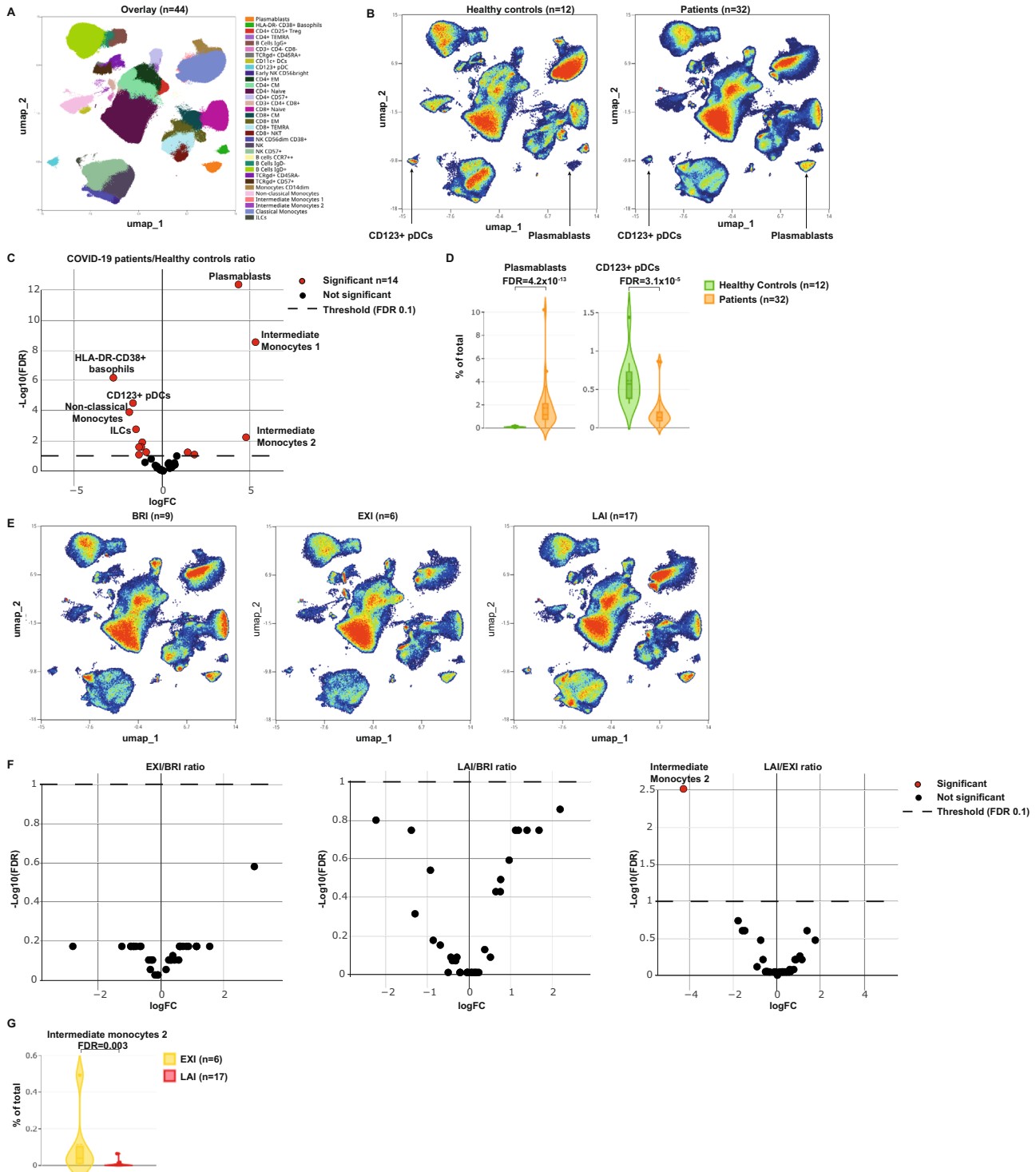

**Fig. 6 High-dimensional flow cytometry reveals distinct immune subset differences between immunotypes. A** Population cluster identification in high-dimensional 40-color flow cytometry data using uniform manifold approximation and projection (UMAP) dimensionality reduction. Visualization is performed on combined COVID-19 patients and healthy control data ($n = 44$). **B** UMAP plots of healthy controls ($n = 12$) and all COVID-19 patients ($n = 32$). Plasmablast and pDC indicated by arrows. **C** Volcano plots of the comparison COVID-19 patients/Healthy controls. Positive fold-changes indicate cellular population increases in patients while negative fold-change indicate decreases in patients. **D** Plasmablast and pDC frequencies in healthy controls and COVID-19 patients show. Violin plots depict the percentage of cells in the total population. False-detection rate (FDR) of population comparisons for plasmablasts is $4.2 \times 10^{-13}$ and for pDC $3.1 \times 10^{-5}$. **E** UMAP plots of the three immunotypes (BRI $n = 9$, EXI $n = 6$, LAI $n = 17$). **F** Volcano plots of the pairwise comparison of immunotypes. Positive fold-change indicates increase in numerator of ratio, while negative fold-change indicates decreases. All patients are from Rotterdam cohort. **G** Intermediate monocyte 2 frequencies in EXI and LAI patients show. Violin plots depict the percentage of cells in the total population. The violin shapes were defined by kernel density estimation the boxes define the Q1 to Q3 range while box whiskers indicate min/max. The mean and median are indicated by dotted and solid middle lines, respectively. FDR of population comparison is 0.003.

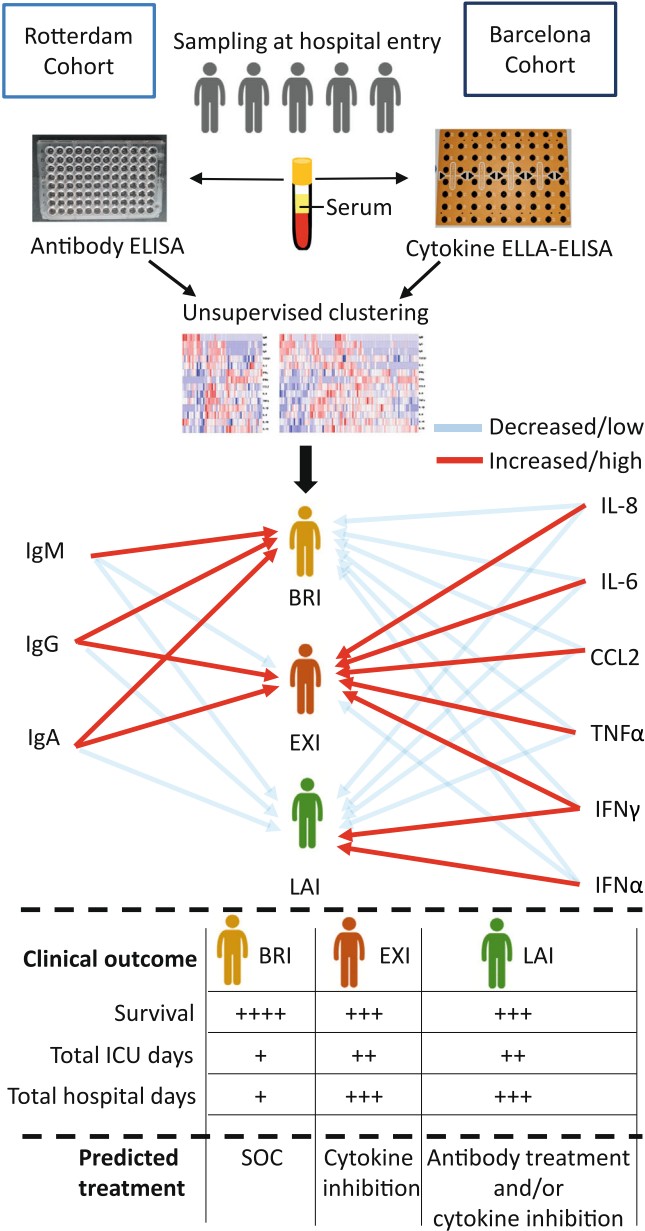

**Fig. 7 Unsupervised clustering defines three immunotypes at hospital entry that predict clinical outcome.** Anti-SARS-CoV2 nucleocapsid antibodies (IgM, IgG, IgA) and the most prominent anti- and pro-inflammatory cytokines out of $n = 14$ shown. BRI: balanced response immunotype; EXI: excessive inflammation immunotype; LAI: low antibody immunotype; SOC: standard of care.

each other and this may explain the relatively small benefit and mixed results of anti-IL-6 antibody treatment[29–31]. Using immunotypes to predict who will develop a hyperinflammatory syndrome[32,33] may enable pre-emptively blocking key drivers of the pro-inflammatory cytokine network. This would provide the maximal clinical benefit as dampening of the cytokine storm is achieved when anti-cytokine treatment is applied early[34–37]. The potential benefit of prophylactic anti-cytokine treatment for COVID-19 is also suggested from autoimmune patients in which anti-cytokine therapy associates with milder COVID-19 disease[38–40]. Longitudinal studies have shown anti-spike IgA and antibody affinity can correlate with disease severity[41,42] and this could be driven by persistent antigen. In contrast, our study that samples patients at hospital entry, reveals that patients with

muted antibody responses do worse clinically in the future. Such patients may benefit the most from anti-SARS-CoV-2 antibody therapy. Although convalescent plasma therapy has failed to show clear benefit in COVID-19[43,44], this could be due to most patients already having antibodies. Treatment with convalescent plasma of specific populations at risk very early in the disease course have suggested benefit in terms of preventing clinical deterioration[45]. Therefore, patients within the LAI group, that have no or low anti-SARS-CoV-2 antibodies may stand to benefit from high neutralizing antibody titer plasma or monoclonal antibodies. Finally, the observation that very few patients in the BRI immunotype progressed to more severe disease could help hospitals overwhelmed with COVID-19 patients to triage which patients could be transferred to step-down units outside the hospital. Immunotypes, therefore, could guide the formulation of personalized therapies for COVID-19 patients based on mechanistic evidence.

Our study proposes immunotypes that are free from the presumption that clinical classification should dominate the analysis and could be the basis for prospective studies. From a clinical perspective, it is important to identify at an early stage which patients may progress in disease severity as the clinical condition of COVID-19 patients can rapidly deteriorate within days. Here, we have identified immunotypes that predict disease progression but also shed light on underlying pathways and suggest biomarkers and therapeutic targets before severe COVID-19 develops. Our current study only addressed the immunotypes in the context of acute COVID-19, but it also points towards employing larger cohort studies that may reveal immunotypes that predict long term post-COVID-19 complications. As immunotype identification requires only serum analysis, these immunotypes can be determined rapidly after patient admission and help instruct personalized therapy.

## Methods

**Patients.** Rotterdam cohort samples were collected from patients ($n = 50$) participating in the ConCOVID nationwide multicenter open-label randomized clinical trial in the Netherlands. Inclusion criteria were patient age of at least 18 years, admittance to the hospital for COVID-19 and SARS-CoV-2 genome positive by reverse transcription–polymerase chain reaction (RT-PCR) test in the previous 96 h. All patients entered the study and were sampled at a median of 2 days after hospitalization (Q1, Q3: 1, 3.75 days). No exclusion criteria were applied. Healthy controls were age and sex matched. The study was reviewed and approved by the institutional review board of the Erasmus University Medical Center. Written informed consent was obtained from every patient or legal representative. All samples were processed and frozen within three hours of bleeding.

Barcelona cohort samples and data from patients included in this study were provided by the Hospital Universitari Vall d'Hebron (HUVH) Biobank (PT17/0015/0047), integrated in the Spanish National Biobanks Network. The collection and secondary use of de-identified diagnostic samples and data for research was approved by the Ethics Committee for Research with Medicines of HUVH and consent was waived. Sample and data transfer to Erasmus MC for research purpose was approved by the Ethics Committee for Research with Medicines of HUVH. Patient inclusion criteria were an age of at least 18 years, having a confirmed SARS-CoV-2 genome positive by RT-PCR test and having a serum sample on the day of hospitalization. No exclusion criteria were applied. Clinical severity was classified according to the WHO 8 point COVID-19 disease severity score (at study inclusion for patients and during hospitalization) in which 0 is no clinical or virological evidence of infection, 1 is no limitation of activities, 2 is limitation of activities, 3 is hospitalized, no oxygen, 4 is oxygen by mask or nasal prongs, 5 is non-invasive ventilation or high-flow oxygen, 6 is intubation and mechanical ventilation, 7 is ventilation and additional organ support (vasopressors, renal replacement therapy, ECMO), and 8 is death[9]. Clinical characteristics, laboratory measurements and treatments for both cohorts are shown in Table 1.

Days from symptom onset (DFSO) were determined by asking patients at study inclusion when they experienced the first symptoms of COVID-19 infection. None of the patients were vaccinated against SARS-CoV-2 spike protein.

**Cytokine measurements.** The cytokines of interest (IL-6, TNFα, IL-1β, IL-8, CCL2, IL-18, IL-10, IL-12, IFNγ, IL-5, IL-17A, IL-2, IL-4, IFNα, and TGFβ1) were analyzed using the ELLA Simple Plex system (Protein simple, San Jose, CA). After thawing serum samples on ice, they were centrifuged at 1300 x $g$ for 5 min at room

temperature. Twofold dilutions were prepared in low-protein-binding plates according to the manufacturer's instructions. For TGFβ1, samples were first activated with 1 N HCl and then neutralized with 1.2 N NaOH/0.5 M HEPES. Subsequently, these samples were diluted in a factor 1:15. Diluted samples were loaded into ELLA Simple Plex cartridges and analyzed with the ELLA Simple Plex system.

**Anti-SARS-CoV-2 antibody measurements**. Anti-SARS-CoV-2 IgM, IgG and IgA antibodies against nucleocapsid protein (N-protein) were measured in serum by ELISA using CE-certified COVID-19 IgG ELISA (Tecan, 30177447), COVID-19 IgA ELISA (Tecan, 30177446) and COVID-19 IgM ELISA (Tecan, 30177448) according to the manufacturer's instructions. Positive cutoff for these ELISAs was 11 units. The anti-nucleocapsid ELISA was chosen as it allows the detection of anti-SARS-CoV-2-specific IgM, IgG, and IgA isotypes and circumvents SARS-CoV-2 spike vaccination status in future studies. Serum was tested for the presence of SARS-CoV-2 total Ig and IgM antibodies against spike RBD by ELISA test (Wantai Biological, Beijing) while neutralizing antibodies were measured by performing a plaque-reduction neutralization test (PRNT) with the SARS-CoV-2 virus. Both assays are described in Gharbharan et al.[43].

**High-dimensional flow cytometry**. PBMC from patients and controls of the Rotterdam cohort were stained with a 40-color antibody panel and collected using a 5 laser Aurora spectral flow cytometer (Cytek Biosciences, CA). All samples were stained as described previously[46] with the adaptation of including annexin V to exclude dead cells and all buffers contained all buffers contained 2.5 mM CaCl₂. Cleaning of flow data was performed in SpectroFlo (version 2.2.0) (Supplementary Fig. S4). The unsupervised, and statistical inference portions of the flow cytometry analysis were performed using OMIQ data analysis software(www.omiq.ai). We used the unsupervised analysis methods based on surface markers without any 2D gating we previously employed[46]. The workflow included running flowCut to check for changes in channels over acquisition time, UMAP for dimensionality reduction, flowSOM for clustering, and edgeR for statistical inference. For the statistical comparisons of abundance, the Flow Cytometry Standard (FCS) files were subsampled to ensure the same number of events were included per group (either immunotype or disease state).

**Unsupervised hierarchical clustering and principal component analysis**. Patient's clinical and serum data from the Erasmus MC and Barcelona cohort were loaded into R (v.4.0.4)[47]. Only cytokines positive in >20% of patients in both cohorts were used in the analysis. Standardized scores (z-scores) for each cohort were calculated based on the log10 transformed cytokine and antibody levels measured in serum. Unsupervised hierarchical clustering was performed using Ward's Hierarchical Agglomerative Clustering Method (ward.d2) on the Euclidean distances of the z-scores[48]. The optimal number of clusters for both cohorts was assigned with the NbClust (v1.0.12) package in R[49]. NbClust provided 30 indices that determined the best number of clusters for both datasets and proposed the appropriate number of clusters using the majority rule. The majority rule indicated three clusters as optimal for both cohorts. Subsequently, heatmaps were plotted using the R package pheatmap (v1.0.12). Principal component analysis (PCA) was performed and visualized based on the z-scores using the R function prcomp. The grouping of patients in these different clusters was not influenced in any way by the investigators.

We tested the reliability of our hierarchical clusters by constructing a multinomial logit model[50]. The model describes ln(Prob(EXI) over Prob(BRI)), and ln(Prob(LAI) over Prob(BRI)) in relation with the variables used in building trees in Fig. 2. For a better coherence with the method used in building the three categories, we log-transformed the 14 cytokines and antibody variables as we used to construct the trees in Fig. 2, while removing the patients with no CCL2 data (BRI = 32, EXI = 46, LAI = 58). The model was 100% accurate in predicting the immunotypes based on the input data. We estimated the stratification accuracy from a predictive perspective cross-validation by splitting the dataset into a test and training set, increasing test percentages from 5%, 10%, 15%, 20%, and 25% using 150 iterations.

**Statistical analysis**. Statistical analysis was performed using R (v.4.0.4)[47]. Normality of the patients' data was tested using a Shapiro–Wilk normality test. Statistically significant differences between immunotypes were calculated using two-sided multiple Students t-tests for patient's age and Wilcoxon rank-sum tests for all the other variables. Subsequently, p-values were adjusted using Bonferroni correction. Adjusted p-values lower than 0.05 were considered statistically significant. The number of asterisks indicate the level of significance of p-adjusted values: *p < 0.05, **p < 0.01, ***p < 0.001, and ****p < 0.0001. Non-significant results are not shown in figures. Correlations coefficients between the variables in each cohort were calculated using a Spearman's rank correlation coefficient and presented in correlograms and DFSO correlations. A correlation was considered statistically significant if the p-value was lower than 0.05. Non-parametric local polynomial regression lines were plotted using the locally estimated scatterplot smoothing (LOESS) method. We applied a Uni- and Multivariate analyses next to our Wilcoxon rank-sum/Mann–Whitney U-test that directly compared cytokine, antibodies and other variables in this study. We applied these analyses of the log-

transformed data using a univariate binary logistic regression of BRI vs. EXI, BRI vs. LAI and EXI vs. LAI. In addition, we applied on the same variables a multivariate analysis with gender, age, and ln(DFSO + 1) as additional variables. Two patients with no CCL2 measurement were dismissed from analyses while two patients with no DFSO indication were removed in the multivariate analyses. Wilcoxon rank-sum/Mann–Whitney U-test p-values were adjusted using Bonferroni correction on the three comparisons.

**Reporting summary**. Further information on research design is available in the Nature Research Reporting Summary linked to this article.

## Data availability
The data used in this study are available on request from the corresponding author PDK. The data are not publicly available due to participant privacy/consent. Source data are provided with this paper.

## Code availability
The R-code used to cluster and the statistical analysis the cytokines, antibodies and clinical data can be freely downloaded from https://bitbucket.org/immunology-emc/covid_severity_publication/src/master/.

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

## Acknowledgements

We would like to thank all the patients who participated in the study. We thank all physicians, medical students, study nurses and trial coordinators in the participating centers. We would like to thank Bart Haagmans and Corine Geurts van Kessel from the Viroscience department, Erasmus MC for serum anti-RDB and PRNT50 antibody data of CONCOVID trial[43]. We want to particularly acknowledge the patients and the HUVH Biobank (PT17/0015/0047) integrated in the Spanish National Biobanks Network for their collaboration. Thanks to BMW, the Netherlands for providing two free-of-charge vehicles for sample collection. This work was supported by Health Holland LSHM20056 grant (PDK), in part from the European Union's Horizon 2020 research and innovation program under grant agreement No 779295 (PDK), in part supported by the Erasmus foundation (BJAR), grant PI20/00416 from the Instituto de Salud Carlos III (RPB) and the European Regional Development Fund (ERDF) (RPB).

## Author contributions

Y.M.M., T.J.S., R.R., M.W.J.S., D.A.S., C.H.K., M.D.C.E., M.v.M., I.B.H., M.Z., L.L., H.d.W., C.A.O., M.E.P.W., T.M.A., D.A.L., T.v.W., M.C.J., G.K., S.v.B., C.R., B.J.A.R., M.H.G., R.P.B., contributed to data acquisition and analysis. D.G.B.G., G.J.L., and H.J.G.v.d.W. performed machine learning, bioinformatics, and statistical analysis. P.D.K. conceived and designed the work, and wrote the manuscript. All authors reviewed and edited the manuscript.

## Competing interests

The authors declare no competing interests.
