## [Peer Review File · Nature Communications]

Stratification of hospitalised COVID-19 patients into clinical severity progression groups by immuno-phenotyping and machine learningEditorial Note: This manuscript has been previously reviewed at another journal that is not operating a transparent peer review scheme. This document only contains reviewer comments and rebuttal letters for versions considered at *Nature Communications*.

REVIEWER COMMENTS

Reviewer #1 (Remarks to the Author):

This study is one of the first to characterize immunotypes for COVID-19 in a manner independent of disease severity. The assumption is that the disease kinetics can vary between patients leading to different disease progressions on different timescales. Three immunotypes were discovered based on pro-inflammatory, anti-inflammatory, and anti-viral cytokines and antibody responses: balanced response immunotype (BRI), excessive inflammation immunotype (EXI), and low antibody immunotype (LAI). The results were consistent across two locations. BRI patients had the best outcome, and EXI and LAI showed two distinct progressions with worse outcomes. These immunotypes could not be explained by age or disease duration. Although differences between COVID-19 patients and healthy patients are observed, the predicted progression of the disease can only be subtly observed by monitoring the immune cell changes in the blood of patients, mainly via intermediate monocytes 2 between EXI and LAI. Cytokine levels are necessary as well to differentiate the immunotypes. These immunotypes point to different treatment strategies.

I believe this is a solid manuscript that has been reviewed and revised adequately in the previous Nature submission. It's a great example of how machine learning can potentially be used to provide some insights into phenotypes of diseases. The manuscript reads well and the motivation, methods, and results overall read well. The figures also provide many ways of viewing the results. The authors have provided a thorough and robust way to deal with the relatively small sample size of their study, through careful sensitivity analysis for the accuracy vs test size, which increases confidence in the paper's results. I did not have any major issues in the previous reviewing round and throughout this transfer and I believe it will be a good paper for this journal.

Reviewer #2 (Remarks to the Author):

Mueller et al, performed immunophenotyping of adult COVID patients and discussed their findings to predict COVID-19 clinical severity.

This manuscript is missing important critical information, and there are several biases in the grouping of patients that impacts the conclusion drawn by this study, that precludes publication in its current form.

MAJOR COMMENTS:

1. There seems to be an arbitrary reasoning for defining different immunophenotypes and is confusing. Based on antibody and cytokine profile, first 3 patients in EXI group in primary cohort should be classified in LAI (Fig. 1A). This becomes even more subjective (illogical) for the Barcelona validation cohort wherein approx. 1/3rd of the EXI group should belong to LAI group. The individuals should be accurately grouped between EXI and LAI and data should be reanalyzed to provide reasonable conclusions.
2. It's easy to immunologically differentiate between healthy individuals and COVID-19 patients, but healthy individuals are not the appropriate controls for the COVID-19 disease. Please analyze and include data for age-matched hospitalized patients that have other phenotypically similar disease states to serve as alike-disease matched controls to identify differences in immune responses unique to SARS-CoV-2 infection to make meaningful conclusions.
3. Clinically relevant antibodies target SARS-CoV-2 spike for vaccine and therapeutics. Its surprising that authors only analyzed anti-Nucleocapsid response and by ELISA. Multiple studies have shown that higher disease severity have more robust anti-spike and neutralizing antibody titers than milder disease patients. Its critically important to analyze both spike- binding IgM, IgG and IgA as well as SARS-CoV-2 neutralizing antibodies and include this data in the comparative immunophenotyping analysis in the 2 study cohorts.
4. There was significant difference in gender between the 3 immunophenotypes (Table S2). More males in EXI and LAI compared to BRI. Whole data could be explained by higher propensity of

females in the BRI. Males have been observed to higher proinflammatory cytokines and disease severity in previous COVID-19 studies. Please discuss. The statistical analysis in various figures and table should be reanalyzed by normalizing for age, gender and DFSO using multivariate analysis.

5. There is lot of overlap in the data and significance is driven by few outliers especially in EXI group for e.g. in the EXI group Fig. 3A. and may be just driven by these few outliers in EXI/LAI compared with BRI. Please address this caveat by reanalyzing the data and discuss further.

6. Manuscript should include longitudinal analysis of samples from both early/acute phase and later time-point samples and their correlation with disease outcome to confirm the conclusions in the study. Even though authors stress that DFSO does not impact their study outcome, the data in figures and supplementary tables clearly show an impact of DFSO is linked to both cytokine and antibody responses. Several studies have provided evidence of DFSO timing on innate and adaptive immune response. So, at the least, authors should provide the data for samples analyzed from the same time post-onset (DFSO) across all study participants.

7. Similar to other studies, this manuscript shows that the IL-6, IL-8 and IL-10 are main markers of disease severity and level of these cytokines can easily explain disease outcome among immunotypes (Fig. 2). So that brings into question the rationale of the whole study. Please compare and discuss.

8. Anti-N IgM seems to be only antibody difference between BRI and EXI/LAI in Fig. 1A. Does lower class-switching of IgM correlate with disease severity or outcome?

9. CRP and ferritin levels demarcate the groups. What is impact of convalescent plasma treatment on the immunophenotype in these BRI/EXI/LAI cohorts?

10. Authors suggest that antibody treatment may help clinical outcome in some patients. However, large-scale randomized trials of convalescent plasma and IVIG treatment studies have failed to show an impact. Please discuss this further in context of your findings.

11. Authors conclude that this immunophenotyping strategy can be applied rapidly in clinic for a personalized treatment strategy. It's unclear, how this extensive analysis and tedious approach can be practically applied in clinical setting where the patient deteriorates in matter of days? During the pandemic, with shortage of resources, clinical staff, and lack of scientist and machine learning tools in a hospital that are proposed here, this approach is simply impractical to help point-of-care in any clinical practice even in most advanced countries.

12. What could be prognostic markers that define appropriate therapeutic intervention? It would help to add a figure schematic to summarize the findings.

13. Methods are missing critical information as how the assays were performed. Please provide detailed methods. For eg, how the plasmablast assays were performed?

REPLY TO REVIEWERS

We thank the reviewers for their constructive critique. We have answered the questions/remarks by Reviewer #2 point-by-point below (in blue text). We have highlighted in yellow, the text that addresses the comments by reviewer #2 and also added page and line numbers (empty lines were counted).

Response to Reviewer #1:

We thank the reviewer for finding our study solid and a great example of how to apply machine learning to decipher phenotypes of diseases.

Response to Reviewer #2:

We thank the reviewer for his/her comments and suggestions to improve the manuscript. A point-by-point response to the reviewer's comments follows below.

General comment: This manuscript is missing important critical information, and there are several biases in the grouping of patients that impacts the conclusion drawn by this study, that precludes publication in its current form.

Reply:

We have now added the data requested. In reference to the bias in grouping, we would like to stress that we do not analyze patients after first classifying them into disease severity groups, which is the approach the vast majority of studies have taken. We analyze our patients agnostically to clinical status and data. In addition, the manuscript includes analysis that address the age and gender bias questions raised (see below).

MAJOR COMMENTS:

1. There seems to be an arbitrary reasoning for defining different immunophenotypes and is confusing. Based on antibody and cytokine profile, first 3 patients in EXI group in primary cohort should be classified in LAI (Fig. 1A). This becomes even more subjective (illogical) for the Barcelona validation cohort wherein approx. 1/3rd of the EXI group should belong to LAI group. The individuals should be accurately grouped between EXI and LAI and data should be reanalyzed to provide reasonable conclusions.

Reply: We would like to point out that we, the investigators, do not assign patients to groups, hence there is no arbitrary reasoning (actually there is no reasoning on behalf of the investigators). The clustering of patients was done exclusively in an unsupervised manner by machine learning. At first glance, there are indeed some patients who appear more similar to other immunotypes than the one assigned to by machine learning. An example is low antibody patients in EXI. However, machine learning classifies these patients from the synthesis of markers which go beyond antibodies and includes pro-inflammatory and anti-inflammatory responses and interferon response. For example, for the Barcelona samples, the 1/3 of patients the reviewer considers misclassified by antibodies, clearly lack IFN responses while having high pro-inflammatory cytokines (CCL2, IL-6, IL-8, etc). This makes them assemble with the EXI patients that have high proinflammatory cytokines and low IFN responses despite having low antibodies.

We have stressed in the manuscript that the investigators were not involved in grouping patients and that the immunotype classification is solely due to unsupervised clustering (page 3, line 8).

In addition to having used two cohorts for the analysis (discovery and validation cohorts), we have further tested the robustness of the patient grouping using different training sets and further confirmed the robustness of the immunotypes (page 3, line 18).

2. It's easy to immunologically differentiate between healthy individuals and COVID-19 patients, but healthy individuals are not the appropriate controls for the COVID-19 disease. Please analyze and include data for age-matched hospitalized patients that have other phenotypically similar disease states to serve as alike-disease matched controls to identify differences in immune responses unique to SARS-CoV-2 infection to make meaningful conclusions.

Reply: The goal of our study was not to identify COVID-19 specific biomarkers. Therefore, our purpose was not to describe the differences between COVID-19 patients and healthy controls or non-COVID disease controls. Our goal was to identify biomarkers or identify patient groups/characteristics that can facilitate biomarker discovery that predict disease severity progression and outcome within COVID-19 patients. We do not seek to prove that COVID-19 differs from other respiratory infection (which it may) but rather to discover immunotypes or biomarkers within COVID-19 patients at hospital admission. The healthy controls included in our study serve to provide a normal range. We have now stressed in the manuscript the goal of our study (page 6, line 2).

3. Clinically relevant antibodies target SARS-CoV-2 spike for vaccine and therapeutics. Its surprising that authros only analyzed anti-Nucleocapsid response and by ELISA. Multiple studies have shown that higher disease severity have more robust anti-spike and neutralizing antibody titers than milder disease patients. Longitudinal studies have shown correlation of anti-spike IgA and antibody affinity with disease severity (PMID: 33674317, PMID: 33619281). Its critically important to analyze both spike- binding IgM, IgG and IgA as well as SARS-CoV-2 neutralizing antibodies and include this data in the comparative immunophenotyping analysis in the 2 study cohorts.

Reply: The reviewer is correct that previous studies have indicated that antibody levels or affinity can correlate with severity. An important difference from previous studies is that our samples are collected at hospital entry when most patients have mild to moderate disease. This could explain differences in antibody associations because most of our samples are before patients develop severe disease. We now discuss this important issue in the discussion and reference the two important studies suggested (page 8, line 10, Ref 41 and 42). Our goal is to show if the adaptive antibody response differs in some patients (impaired or delayed?) and not to show a correlation between disease severity and antibody response and/or neutralizing antibodies which may be true at later time points and after days of hospitalization.

We picked the anti-nucleocapsid ELISA for a number of reasons. Firstly, we chose the anti-nucleocapsid ELISA as it allowed us to analyze IgM, IgG, and IgA whereas other ELISA such as the anti-spike RDB ELISA from Wantai, at the time of this study, allowed only the analysis of total Ig and IgM. Secondly, the anti-nucleocapsid ELISA from Tecan, we used in this study, was already CE-certified at study time point, which was not the case for the Wantai anti-Spike ELISA (CE declaration of Conformity January 2021). We have now indicated the above in the material and method (page 9, line 33) as well as the result section (page 3, line 26).

Thirdly, anti-NC antibodies are an excellent surrogate for the overall anti-viral antibody response. We now report in supplementary figure 2, that anti-nucleocapsid antibodies strongly correlate with the anti-spike

RDB total Ig and plaque reduction neutralization assay (PRNT50) titers for SARS-CoV-2 neutralizing antibodies in our Rotterdam cohort (we did not have material to test the Barcelona samples). We have also present this data in the results section (page 3, line 37).

A final reason was that vaccination status does not affect directly the anti-nucleocapsid response, something important for future studies, as more individuals are being vaccinated against spike protein. We mention this in the material and method (page 9, line 33). To clarify vaccination status, we now mention on page 9 line 19 that none of the patients are vaccinated.

4. There was significant difference in gender between the 3 immunophenotypes (Table S2). More males in EXI and LAI compared to BRI. Whole data could be explained by higher propensity of females in the BRI. Males have been observed to higher proinflammatory cytokines and disease severity in previous COVID-19 studies. Please discuss. The statistical analysis in various figures and table should be reanalyzed by normalizing for age, gender and DFSO using multivariate analysis.

Reply: Reviewer #2 is correct that gender is a factor, with males being at higher risk to belong to EXI immunotype. However, gender alone does not completely define immunotypes. We have included a new figure showing the sex distribution between the three different immunotypes (Figure 3F). Although there is a significant difference in sex distribution between BRI and EXI, no significant difference was found between BRI and LAI as well as EXI and LAI. This indicates that although gender plays a role, it is not sufficient to predict the immunotypes.

We have applied a Uni- and Multivariate analyses next to our Wilcoxon rank-sum/Mann-Whitney *U*-test that directly compared cytokine, antibodies and other variables in this study. We applied these analyses on the log transformed data using a binary logistic regression of BRI vs EXI, BRI vs LAI and EXI vs LAI. Moreover, we applied on the same variables a multivariate analysis with Gender, Age and ln(DFSO+1) as additional variables. No striking difference between the non-parametric and the parametric tests and between the uni- and multivariate analysis were detected. The results of the analyses can be found in supplementary table 3. These results strikingly show that BRI and LAI do not differ by most “inflammatory” markers yet they have a very different clinical course after hospital entry. Sex, age or DFSO do not affect these findings.

These results have been added to the manuscript (results: page 4, line 23; discussion page 7, line 4) and shown in supplementary table 3.

5. There is lot of overlap in the data and significance is driven by few outliers especially in EXI group for e.g. in the EXI group Fig. 3A. and may be just driven by these few outliers in EXI/LAI compared with BRI. Please address this caveat by reanalyzing the data and discuss further.

Reply: We, respectfully, do not agree that our data and significance is driven by outliers. As reviewer #2 correctly observed, there are indeed outliers present in some of the data. However, because of this we have chosen to perform non-parametric analysis which leads to outliers having a minimal effect. Our supplementary table 3 that includes univariate and multivariate regression analysis further strengthens our conclusions.

6. Manuscript should include longitudinal analysis of samples from both early/acute phase and later time-point samples and their correlation with disease outcome to confirm the conclusions in the study. Even

though authors stress that DFSO does not impact their study outcome, the data in figures and supplementary tables clearly show an impact of DFSO is linked to both cytokine and antibody responses. Several studies have provided evidence of DFSO timing on innate and adaptive immune response. So, at the least, authors should provide the data for samples analyzed from the same time post-onset (DFSO) across all study participants.

Reply: Reviewer #2 asks for a longitudinal study of early and late disease. However, the scope of our study was not to identify correlates between early and late disease. Instead, we only focus on patients at hospital entry to find chronologically predictive markers of disease outcome. The reviewer would agree that this is when biomarkers are most needed and clinical decisions need to be made. We stress this in abstract and manuscript (abstract, page 1 line 25; introduction: page 2, line 16; results: page 3 line 4; discussion: page 6, line 2).

The reviewer is correct that LAI group has lower DFSO compared to BRI and EXI (we show this in Fig 4A). However, we disagree with the conclusion that DFSO is linked to both antibody and cytokine responses. As we show in the non-parametric local polynomial regression 4B and 4C, regardless of DFSO the immunotypes retain the distinct characteristics. This is even more clear in Fig 4D, where we breakdown DFSO segments and for example for DFSO 8-12 days, LAI continues to have low antibodies compared to the other 2 immunotypes. As mentioned above, any role of DFSO in driving our cytokine and antibody data is further excluded by our new univariate and multivariate regression analysis that takes into account DFSO. Please see Supplementary Table 3 and text in results page 4, line 36 and discussion page 7, line 4.

We now also examined whether DFSO correlate with days in hospital and days in the ICU, and found no correlation between DFSO at entering the study and length of stay in the hospital and ICU. This finding further stresses that DFSO is not predictive for disease severity. We report this now in the results on page 5, line 5.

7. Similar to other studies, this manuscript shows that the IL-6, IL-8 and IL-10 are main markers of disease severity and level of these cytokines can easily explain disease outcome among immunotypes (Fig. 2). So that brings into question the rationale of the whole study. Please compare and discuss.

Reply: We respectfully disagree that the level of inflammatory cytokines can easily explain disease outcome among immunotypes. Disease severity and worse outcome in EXI and LAI immunotypes compared to BRI appear to associate with higher levels of IL-6 and IL-10 but not IL-8, as shown in Fig 2A. However, using for example IL-6 or IL-8 in the 10-100pg/ml range, there is a very large overlap between BRI (which have a mild clinical course) and EXI/LAI (see Fig 2A). We believe the reviewer would agree that it would be impossible to distinguish these BRI patients and make clinical decisions based on these markers. Importantly, the immunotypes we identify also inform on different underlying mechanisms and possible treatments. For examples, EXI and LAI differ significantly in their antibody response. This has also implications for therapy, since pro-inflammatory cytokines are significantly higher in EXI compared to LAI. We discuss this on page 6, lines 16 and 37.

8. Anti-N IgM seems to be only antibody difference between BRI and EXI/LAI in Fig. 1A. Does lower class-switching of IgM correlate with disease severity or outcome?

Reply: Although it appears that IgM is higher in BRI versus EXI in Fig 1A when the data were statistically analyzed in Fig 2B, we find that IgM levels are not significantly different between BRI and EXI. LAI does

however have reduced IgM (as it does for IgG and IgA). Therefore, there is no evidence in our data of lower class-switching of IgM. Lower class-switching IgM would lead to higher IgM with lower levels of IgG or IgA. The vast majority of our patients have IgG or IgA when IgM is present.

9. CRP and ferritin levels demarcate the groups. What is impact of convalescent plasma treatment on the immunophenotype in these BRI/EXI/LAI cohorts?

Reply: We respectfully disagree with Reviewer #2 that CRP and ferritin levels demarcate the groups. CRP, ferritin, D-dimer and LDH are not significantly different between BRI and LAI, two immunotypes that have very different clinical outcomes. Only EXI has these inflammatory markers increased (Fig 5). Thus predicting who will do worse at hospital entry using these markers would capture only a subset of patients.

We have previously shown that convalescent plasma has no effect on disease outcome (PMID: 34045486). This is not surprising as the amount of antibody transferred with convalescent plasma is quite low. We stress that our samples were obtained and immunotypes identified at hospital entry, before any treatments. We have now included further discussion on convalescent plasma or monoclonal antibodies (page 8, line 13).

10. Authors suggest that antibody treatment may help clinical outcome in some patients. However, large-scale randomized trials of convalescent plasma and IVIG treatment studies have failed to show an impact. Please discuss this further in context of your findings.

Reply: Reviewer #2 is correct that most convalescence or IVIG studies have disappointedly failed. This however, may be a problem of clinical study design, as we show that most patients already have good antibody responses. Providing 400 ml of plasma with 1/640 titer antibodies results in a small addition of antibodies to those already in these patients. Perhaps, selecting patients such as LAI would show benefits. It is indeed the goal of this study to define who would benefit from specific treatments such as antibodies (monoclonal or plasma). Our study indicates that high dose antibody may be most beneficial for LAI patients, as these have low and delayed antibody responses. Our study also shows that BRI and EXI patients already have an antibody response, therefore treating with antibodies may not have an effect. We comment on convalescent plasma in the discussion (page 8, line 13).

11. Authors conclude that this immunophenotyping strategy can be applied rapidly in clinic for a personalized treatment strategy. It's unclear, how this extensive analysis and tedious approach can be practically applied in clinical setting where the patient deteriorates in matter of days? During the pandemic, with shortage of resources, clinical staff, and lack of scientist and machine learning tools in a hospital that are proposed here, this approach is simply impractical to help point-of-care in any clinical practice even in most advanced countries.

Reply: Reviewer #2 touches upon an issue that we consider very important in biomarker discovery and that is ease and speed of measurement. The immunotypes discovered, may also facilitate the prioritizing of subsets of markers that could be used for the development of algorithms. The study presented in this manuscript is the first step to define different immunotypes that are clinically meaningful at hospital entry. We purposefully employed assays that are not laborious or technically demanding and can be employed in a routine diagnostic laboratory. The antibody ELISA for anti-NC can be done robotically by a Phadia system while the cytokine ELLA capillary ELISA assay is highly reproducible, and is hands-free after

sample application to multi-cytokine cartridge, taking only 2 hours for results. Our ultimate goal is to develop algorithms with only a few of these measurements included. This could even result in a point of care assay as both assays are ELISA based. Our multinomial logistic regression approach is a first step. We are currently, seeking funds for a much larger cohort of close to 1000 patients that would allow the testing of predictive algorithms.

12. What could be prognostic markers that define appropriate therapeutic intervention? It would help to add a figure schematic to summarize the findings.

Reply: We agree with the reviewer that the ultimate goal is to find a small number of prognostic markers which would guide the treatment of COVID-19 patients at hospital admission. As outlined above, this study is a first step to defining these biomarkers.

We have included a summary figure (figure 7) to convey the major strategy and findings of the study and hope that the reviewer finds this figure helpful.

13. Methods are missing critical information as how the assays were performed. Please provide detailed methods. For eg, how the plasmablast assays were performed?

Reply: We apologize for the omissions. The cytokine and antibody assays used are commercially available and done according to manufacturer's instructions. Bioinformatic analysis is reported in detail and code is provided. For the high dimensional flow cytometry, we list all the antibodies used in the attached metadata. For the markers used for population definitions, we cite in the methods reference 46 (Park LM et al; PMID: 32830910) that describes in details the method, markers and population definitions used in our study.

For defining plasmablasts, no specific assay was used. High dimensional flow cytometry was employed and these cells were defined by flow cytometry (CD45+CD19+CD20-CD27+CD38+IgD-) as outlined in reference 46 (Park LM et al; PMID: 32830910).

REVIEWERS' COMMENTS

Reviewer #2 (Remarks to the Author):

Authors have responded to all the comments satisfactorily.

The revised manuscript has improved considerably and new figure 7 summarizes the study very well. Kudos to the authors.